*GigaScience*, 2024, **13**, 1–15

Data Note

# "UDE DIATOMS in the Wild 2024": a new image dataset of freshwater diatoms for training deep learning models

Aishwarya Venkataramanan [1,2,3,‡], Michael Kloster [4,*,‡], Andrea Burfeid-Castellanos [4], Mimoza Dani[4], Ntambwe A. S. Mayombo [4], Danijela Vidakovic [4,5], Daniel Langenkämper [6], Mingkun Tan [6], Cedric Pradalier [2], Tim Nattkemper [6], Martin Laviale [1,3], and Bánk Beszteri [4]

[1]Université de Lorraine, CNRS, LIEC, F-57000 Metz, France
[2]Georgia Tech Europe, CNRS IRL 2958, F-57000 Metz, France
[3]LTSER-"Zone Atelier Moselle", F-57000 Metz, France
[4]Phycology Group, Faculty of Biology, University of Duisburg-Essen, 45141 Essen, Germany
[5]Institute of Chemistry, Technology and Metallurgy, University of Belgrade, National Institute of the Republic of Serbia, 11000 Belgrade, Serbia
[6]Biodata Mining Group, Faculty of Technology, Bielefeld University, 33615 Bielefeld, Germany
*Correspondence address. Michael Kloster, University of Duisburg-Essen, Universitätsstr. 2, 45141 Essen. E-mail: michael.kloster@uni-due.de
‡Equal contribution

## Abstract

**Background:** Diatoms are microalgae with finely ornamented microscopic silica shells. Their taxonomic identification by light microscopy is routinely used as part of community ecological research as well as ecological status assessment of aquatic ecosystems, and a need for digitalization of these methods has long been recognized. Alongside their high taxonomic and morphological diversity, several other factors make diatoms highly challenging for deep learning–based identification using light microscopy images. These include (i) an unusually high intraclass variability combined with small between-class differences, (ii) a rather different visual appearance of specimens depending on their orientation on the microscope slide, and (iii) the limited availability of diatom experts for accurate taxonomic annotation.

**Findings:** We present the largest diatom image dataset thus far, aimed at facilitating the application and benchmarking of innovative deep learning methods to the diatom identification problem on realistic research data, "UDE DIATOMS in the Wild 2024." The dataset contains 83,570 images of 611 diatom taxa, 101 of which are represented by at least 100 examples and 144 by at least 50 examples each. We showcase this dataset in 2 innovative analyses that address individual aspects of the above challenges using subclustering to deal with visually heterogeneous classes, out-of-distribution sample detection, and self-supervised learning.

**Conclusions:** The problem of image-based identification of diatoms is both important for environmental research and challenging from the machine learning perspective. By making available the so far largest image dataset, accompanied by innovative analyses, this contribution will facilitate addressing these points by the scientific community.

**Keywords:** diatom, light microscopy, digital imaging, slide scanning, aquatic ecology, deep learning, out-of-distribution detection, self-supervised learning

# Data Description

## Context

Diatoms, in systematics mostly referred to as Bacillariophyta [1], though recently also as Diatomea [2], a subgroup of the Stramenopiles under the supergroup TSAR [3], are an ecologically important group of single-celled, chlorophyll-*a* and -*c* containing microalgae. One of their main characteristic cellular features is their production of peculiarly shaped and patterned cell walls, termed frustules, that are composed of approximately 90% amorphous silica [4]. Diatoms are ubiquitous and often abundant in diverse aquatic habitats [5, 6] and contribute substantially to numerous important ecosystem functions and biogeochemical cycles [7, 8]. There are an estimated 10,000 to 30,000 described species of diatoms, with many more waiting to be discovered [9, 10]. Although morphology alone is often insufficient to diagnose diatom species [11], the morphologically recognizable diversity of diatoms is probably larger than that of any other protistan group. This morphological diversity has been the basis of a widespread

use of these organisms as ecological and paleo-ecological indicators both in basic and applied research as well as in regulatory biomonitoring [12–15].

A need for a digital transformation of these light microscopic and manual identification methods has long been recognized based on numerous factors. For one, the number of taxonomic experts capable of diatom identification is low and can become a limiting factor when aiming to scale up the spatial-temporal coverage of ecological and biodiversity monitoring [16]. More fundamentally, digital image–based methods have the potential to enable an improved consistency, reproducibility, and objectivity of diatom analysis when compared to identifications performed by human experts directly on a microscope [17, 18]. Experiences indicate that inconsistencies in diatom identification and enumeration can be substantial between different analysts [19–21], which has also been observed for other organismal groups [22, 23]. Over 20 years ago, the ADIAC project developed fundamental approaches for digital imaging and identification [24]. More than

ever, we now need standardized, digital imaging methods combined with digitally supported taxonomic identification to have objective, reproducible, and comparable taxonomic data for rapid processing of large numbers of samples.

With improving possibilities of digital image acquisition and analysis, methods combining medium- to large-scale image data collection with deep neural networks have recently spread rapidly in biodiversity research [25–27], including in the aquatic and microscopic realm [28, 29]. In the case of diatoms, although not yet broadly applied, slide scanning microscopy now provides a possibility of large-scale digital image acquisition suitable for the standard type of diatom preparations [18, 30–34].

High-resolution/high numerical aperture objectives required for diatom analysis offer only a very limited focal depth, so that usually either the valve shape or the valve ornamentation can be seen clearly at a time. Yet, for taxonomic identification, often both of them need to be considered. In manual microscopy, this predicament is solved by focusing up and down through the 3-dimensional structure of a valve until all relevant features have been observed. In previously published diatom image datasets, a single focal plane was preselected by a human expert to expose the most relevant features for each specimen, depending on valve orientation and species. Such a manual approach is not an option in an automated high-throughput processing pipeline, and the problem of finding the optimal focal plane for taxonomic identification of each diatom specimen automatically has not been solved yet. However, automated slide scanning allows one to image a multitude of focal planes and compress their visual information into a single image by focus stacking. This way, all relevant features are contained within a single image, which massively simplifies downstream processing and analysis.

A range of studies have tested the application of deep learning models for diatom object detection [35–40], counting [41], segmentation [42, 43], and classification [30, 44–49]. Here the term *classification*" is used in the machine learning sense (i.e., referring to machine learning models with a categorical target variable); in a biological terminology, it usually addresses taxonomic identification. Diatom localization (using object detection or segmentation models) can now be performed with a high accuracy, even on gigapixel-sized slide scans sometimes termed "virtual slides" [35, 43, 50]; the classification problem (taxon identification), however, remains highly challenging.

Several factors make the diatom classification problem particularly challenging from the machine learning or computer vision perspective. The high number of observed species is a challenge by itself: even when focusing on a local or regional flora, the number of diatom species often lies in the hundreds. In geographically more extended settings, the number of species can quickly reach thousands [51]. According to published experiences, between 50 and 100 examples (ideally, more) per taxon are required for deep learning model training to reach satisfying classifier performances [45, 46]. Collecting and annotating so many images using a manual approach (as done so far in most diatom deep learning studies) is highly time-consuming. The problem is exacerbated by the uneven distribution of taxa, leading to most species being encountered comparatively rarely. This is not a peculiarity of diatoms but results from the general ecological phenomenon often termed hollow abundance distributions [52, 53]. From the machine learning perspective, this leads to a class imbalance problem [54–56]. On the practical side, a consequence is that collecting sufficient examples for rare taxa can take orders of magnitude more effort than capturing common taxa.

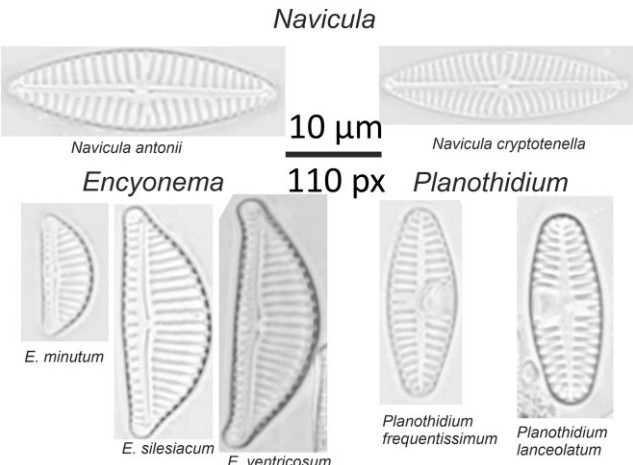

**Figure 1:** Selected examples of diatom specimens. Valvar views from 3 different genera (*Navicula*, *Encyonema*, *Planothidium*), each one with visually highly similar but distinct species.

A further challenging aspect of image-based diatom identification can be summarized as a generally high intraclass (intraspecific) variability often paired with very minute between-class (interspecific) differences (Figs 1 and 2). This is connected to 2 features of the biology of diatoms. First, the diatom life cycle entails a cyclic alteration of size reduction (accompanying vegetative divisions) with size restitution commonly linked with sexual reproduction [57–59]. In taxa with elongated shapes, size diminution is disproportionately faster in the apical (length) than the transapical (width) direction, leading to substantial shape changes during the life cycle (Fig. 2A). Second, environmental effects such as nutrient availability, salinity, or temperature can also lead to morphological variations (ecomorphologies, Fig. 2B; phenotypic plasticity, Fig. 2C). It is common in elongated-shaped diatoms that similar-sized representatives of different closely related species appear visually more similar to each other than to differently sized specimens of the same species [60, 61]. Furthermore, the geometric properties of diatom frustules lead to a further complication in that diatom cells or valves are mostly encountered on microscopic slides in certain viewing angles, mostly in valvar (looking directly onto the valve surface) and/or pleural (looking at the girdle bands) view, with intermediate (tilted) orientations missing or rare (Fig. 2D). This leads to two visually distinct projections representing a single taxon in the light microscopic view. Human analysts learn to interpret and link these views with experience. However, these distinctly different visual appearances probably present a substantial challenge for typical deep learning models by possibly leading to within-class discontinuities in feature space. A further difficulty for algorithms and human analysts alike are taxonomically difficult groups (sometimes referred to as species complexes or *sensu lato* groups), which means that very similar taxa with partially still unresolved taxonomic status show high variability but also intermediate morphologies (Fig. 2E). The existence of heterovalvar diatoms, those that have 2 valves with differences in the ornamentations, can also lead to distinct visual appearances within a taxon (Fig. 2F).

Routine diatom preparations often also contain disturbing background particles such as sediment, clay, small diatom fragments, and sometimes remains of other organisms (e.g., sponge needles, etc.) (Fig. 3). Although careful adjustments during slide preparation can help reduce overlaps of diatom frustules/valves with such disturbing particles and with each other, such

**Figure 2:** Illustrations of some challenges of visual diatom identification. (A) Due to the complex life cycle, the frustule size reduction usually leads to a change in length-to-width ratio, resulting in a different visual appearance. (B) Diatoms can also present ecomorphological variability (i.e., a species can vary in form depending on environmental influences). (C) Diatoms can also vary their morphological traits such as valve ornamentation within a single species (phenotypic plasticity/morphological variability). (D) Valve orientation relative to the imaging optical axis gives different visual appearances: valvar vs. pleural views refer to viewing angles roughly perpendicular to each other and occur most commonly, depending on the species. Intermediate (oblique or tilted) perspectives can usually be found much less frequently. (E) Large diatom species complexes (*sensu lato* taxon groups) can add to morphological variability. One of many examples is *Cocconeis placentula sensu lato*, which includes *Cocconeis placentula*, *Cocconeis euglypta*, *Cocconeis lineata*, and *Cocconeis pseudolineata*. (F) Monoraphid diatoms possess 2 valves with different morphological appearances, where only 1 valve presents a raphe (i.e., an elongated slit), the other not (raphe and rapheless valves, respectively).

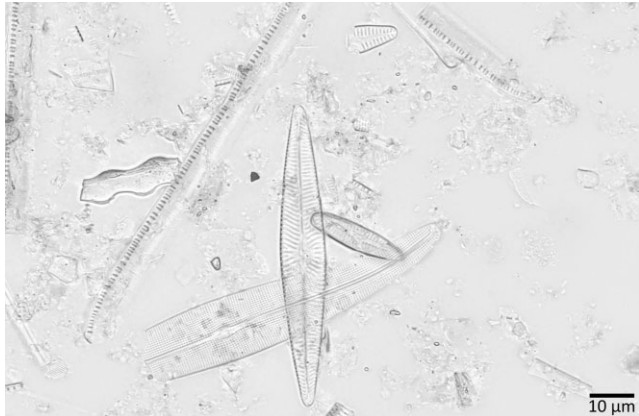

**Figure 3:** Example of a "real-life" diatom preparation. These can, as in this case, contain complex a background (sediment particles and diatom fragments) as well as diatom valves overlapping with each other.

adjustments are rarely performed systematically during routine diatom analysis. This often leads to a situation where diatom frustules touch or overlap with disturbing nondiatom particles or other diatoms, making the visual recognition of taxa more chal-

lenging. Even though these issues are very common, with very few exceptions [43], they are not covered by the currently available diatom datasets (Table 1). Instead of preselecting "clean" examples, we deliberately included such challenging data to get closer to a real-world situation. Even though we cannot offer a solution to all of these problems within the scope of this work, we would like our image dataset to represent a "real-world" difficulty level, which is important for a realistic assessment of the usability of image analysis methods for routine diatom analysis.

Thus, analyses of light microscopic images of diatoms by deep learning are an urgent need for research of ecology and biodiversity, as well as environmental monitoring. Yet, development of the machine learning and computer vision analyses is a challenge. One main obstacle currently slowing the development of the field is the scarcity of datasets that are suitable for training and comparing deep learning models. There are very few publicly available diatom image datasets, and the available ones are mostly too small for training deep learning models. The first published taxonomically annotated light microscopic image dataset addressing a machine learning utilization before the deep learning era came from the ADIAC project [24, 62] and contains ca. 3,400 images representing 328 species. A substantial image dataset known as Aqualitas was assembled a few years ago [45, 61, 63, 64],

**Table 1:** Overview of existing diatom image datasets

| Dataset/Project name | Authors | # of images | # of species | Citation of dataset URL/DOI |
|---|---|---|---|---|
| ADIAC | Du Buf et al. 2000 [24, 62] | 3,400 | 328 | [65] |
| Aqualitas | Bueno et al. 2020 [45, 63, 64] | 10,000 | 100 | [66] |
| Synthetic dataset for diatom automatic detection | Laviale et al. 2023 [35] | 9,230 | 166 | [67] |
| Southern Ocean diatoms (PS79/PS103) | Kloster et al. 2017 [46, 68] | 3,300 | 10 | [69] |
| Kaggle, Diatom Dataset | Gündüz et al. 2022 [70, 71] | 3,027 | 68 | [72] |
| Antarctic Epiphytes | Burfeid-Castellanos et al. 2021 [73, 74] | 18,441 | 120 | [75] |
| UDE PhycoLab Menne | Burfeid-Castellanos et al. 2022 [18, 76] | 8,858 | 161 | [77] |
| Kaggle, scraped from Diatoms.org | Pu et al. 2023 [78] | 7,983 | 1,042 | [79] |
| UDE DIATOMS in the Wild 2024 | This article | 83,570 | 611 | [80, 81] |

covering 100 diatom taxa with about 100 images each. However, the Aqualitas images seem to depict isolated diatom cells, imaged at a single focal plane and containing very little or none of the disturbing factors usually observed in routine preparations (see above). So classification may be considered "too easy" in the context of a nonselective automated imaging workflow. Another dataset was released recently [35], consisting of 9,230 individual images with at least 50 images of 166 diatoms species, which were extracted from pdf versions of publicly available taxonomic atlases [82–84], as well as ca. 600 images of real debris. Another recent study [78] collated images from diatoms.org [85], an online identification aid illustrated by thousands of diatom images, nevertheless still with a relatively low number of examples per species. One dataset containing slightly over 3,300 images of 10 taxa [68] and another one that contains images and segmentation masks for 3,027 diatoms from 68 species [70, 71] are available in public repositories. Two more taxonomically annotated image datasets have been published by Burfeid-Castellanos et al. [18] from a manual digital diatom analysis workflow. These contain 18,441 images of 120 species [73, 74] and 8,858 images of 161 species [76], respectively, averaging 153 and 55 examples per species, although both datasets are imbalanced. The latter two datasets were not explicitly aimed at machine learning utilization and were thus not formatted in a way that would be immediately usable in such a context but could, in principle, also be useful for this purpose. Nevertheless, most published datasets are not ideally suited for deep learning experiments because they are relatively small; Table 1 summarizes basic information on currently available diatom image datasets. We note that for planktonic organisms, a much larger collection of datasets is publicly available, and these were recently reviewed [86].

In this article, we present a novel light microscopic image dataset of freshwater diatoms that (i) is substantially larger than those previously available, (ii) was obtained using a reproducible slide scanning and annotation workflow following standard counting procedures for water quality monitoring [87], (iii) reflects a "real-life" challenge (i.e., it is not limited to manually selected examples that might be biased toward well-recognizable diatoms without, e.g., overlapping debris), (iv) covers the shape as well as the ornamentation of valves/frustules in the same image due to focus stacking, and (v) is publicly available to support customizing and benchmarking deep learning models to this field of application. To highlight the challenging nature of this dataset, as well as to propose possible avenues to address some of these challenges, we provide two deep learning experiments, one addressing out-of-distribution detection and modeling within-class heterogeneity and another one leveraging self-supervised learning to alleviate the need for voluminous labeled training data.

## Methods

### Sampling and preparation

A total of 318 samples of freshwater diatoms were gathered from 15 different localities following standardized methodology,[88] by scraping the biofilm from submerged stones selecting an area of approximately 20 cm$^2$. A total of 5 stones per sampling site were sampled and pooled together. When no stones were available, previously submerged artificial substrates, woody surfaces (epidendron), submerged plants (epiphyton), or sand (epipsammon) were sampled (Appendix 1). The samples were then preserved with molecular-grade ethanol to a final concentration of 75% and stored at $-20°$C.

Diatom preparation followed the hot $H_2O_2$-HCl digestion method [89]. During 5 wash-cycles, the samples were centrifuged at 464 g for 4 minutes, followed by discarding the supernatant and refilling with deionized water. The resulting "clean" sample was oxidized by first treating with 30% hydrogen peroxide ($H_2O_2$), heating up to 90°C for 3–4 hours. After the $H_2O_2$ had evaporated, the samples were left to cool down. Subsequently, 37% hydrogen chloride (HCl) was added to the cooled samples to dissolve the remaining organic matter and carbonates. Finally, after the reaction stopped, the samples were again washed to avoid acid corrosion through prolonged exposure, following the same procedure as during the prewash cycle. After 7 cycles, the sample was suspended in 1 mL deionized water plus 2–3 drops of ethanol or glycerine.

After adding a small amount of 10% ammonium chloride solution to the suspension, it was spread onto coverslips and dried on a heating plate at 350°C. The dried sample on the coverslip was embedded in Naphrax artificial resin with a nominal refractive index of 1.72 (Thorns Biologie Bedarf). The slides were left to harden for one to two weeks before scanning.

### Imaging by slide scanning

The slide preparations were digitized with a VS200 slide scanning microscope (Olympus Europa SE & Co. KG) in bright-field mode using an UPLXAPO60XO 60×/1.42 oil immersion objective. Depending on the preparation's material density, usually 16 or 25 mm$^2$ per slide was scanned in the form of a contiguous rectangular area. To cover the thickness of the sample, mostly 40–85 different focal planes were imaged at a distance of 0.28 μm each; this corresponds to half of the objective's focal depth and warrants

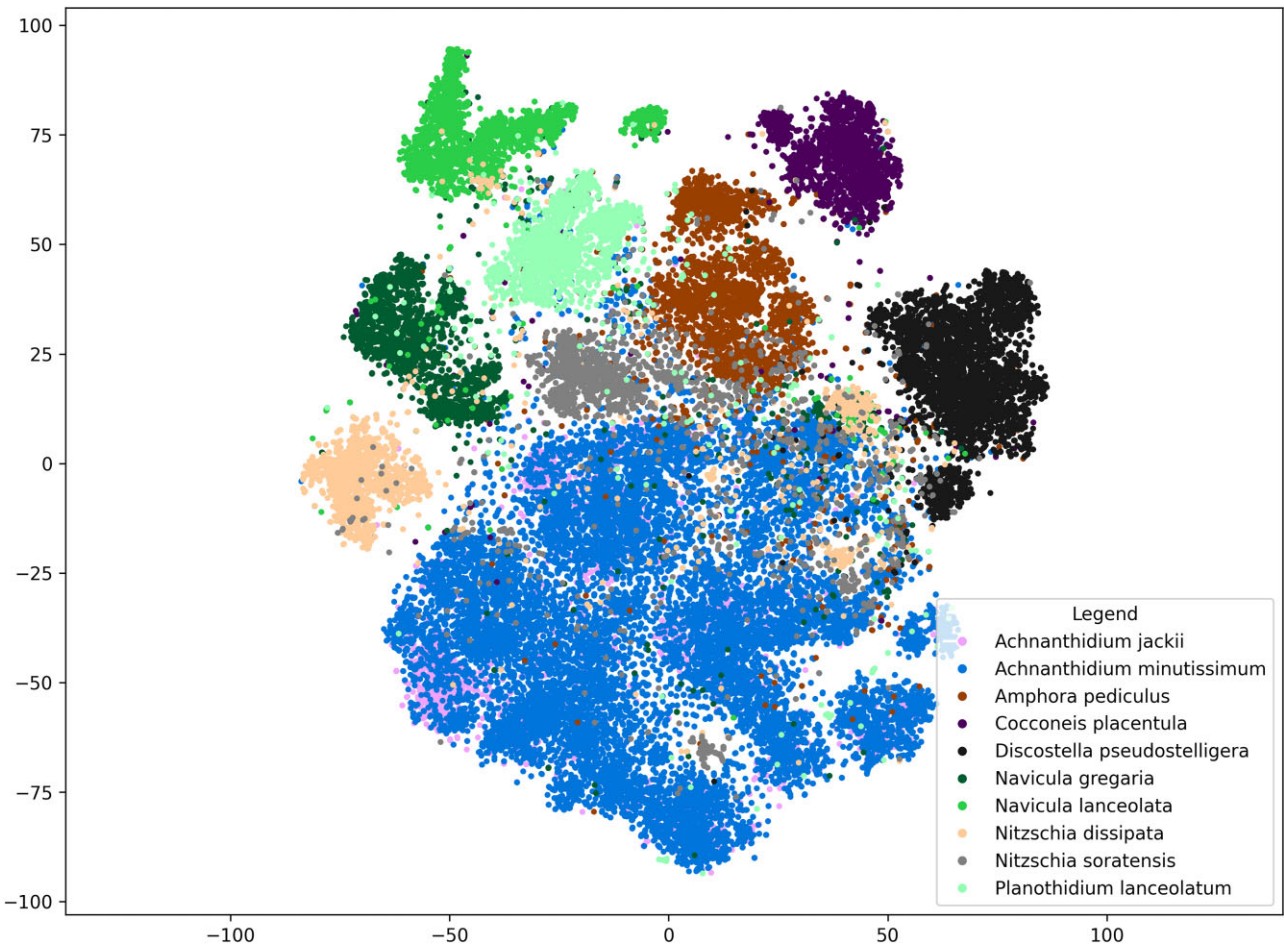

**Figure 4:** The 10 most abundant species visualized in a scatterplot using t-SNE dimensionality reduction. Colors indicate species membership. Each data point depicts one cutout.

that each detail of the valve ornamentation is captured within at least one focal plane. However, due to excessive digital filtering, the VS200 integrated focus stacking tends to suppress fine repetitive structures, which are often essential for diatom identification. To overcome this limitation, we implemented our own postprocessing pipeline utilizing Helicon Focus [90] for focus stacking, the ImageJ plugin "MIST"[91] for position registration of adjacent field-of-view images, and the ImageJ plugin "Grid/collection stitching" [92] for stitching them. Since processed diatom silica does not provide color information, we reduced the 24-bit RGB data to 8-bit grayscale/intensity. A typical slide scan resulted in several gigapixels of image data, divided into subsections of less than 2 gigabyte uncompressed image data, to avoid restrictions of typical image-processing tools and libraries. We refer to such images as "virtual slide images."

## Annotation

Diatoms were annotated using the BIIGLE 2.0 [93] web tool by four annotators (each image was annotated by one of them). Most of the diatom annotations followed the "traditional" microscopy-based workflow as close as possible, screening through a contiguous rectangular area of the virtual slide image. A few samples were processed using random sampling or the so-called lawnmower mode. The latter guides the user over the virtual slide image in a similar serpentine pattern as used during manual counting [93]. As annotation shapes, rectangular bounding boxes,

circles, or polygons roughly outlining the diatom were used. Most annotation shapes were labeled by the specimen's taxonomic name at the species level, some only at genus or down to subspecies level. Taxonomic identification followed standard methodology [87] and was undertaken using general and specific literature [82, 94, 95].

After the identification of at least 400 valves per sample was completed, quality control and consistency checking were executed in a taxon-by-taxon manner with the label review grid overview (LARGO) feature of BIIGLE 2.0 [18].

## Dataset preparation

The annotations were extracted from BIIGLE via the BIIGLE REST API. Subsequently, for each annotation, relevant information was converted into CSV format, and corresponding cutouts from the gigapixel slide scans were generated. Throughout processing, image data were stored in lossless file formats to prevent introducing compression artifacts. We named this dataset "UDE DIATOMS in the Wild 2024" (University of Duisburg-Essen—Digital annotated open-source microscope slide scans from real-world samples, version of 2024).

## Data visualization using dimensionality reduction

To demonstrate the dataset's challenges and to support rendering a mental model of the data distribution, we showcase a

**Table 2:** Metadata files of the dataset

| Column | Content |
| --- | --- |
| annotation_id | Original BIIGLE annotation id (unique ID within the dataset) |
| type | Type of diatom morphology according to the "Diatoms of North America" identification key (https://diatoms.org/morphology) |
| genus | Genus of the annotated specimen |
| species | Species of the annotated specimen ("None" if not identified to species level) |
| subspecies | Historical subspecies or species complex of the annotated specimen, might be shifted to a different species in the near future ("None" if not identified to subspecies level) |
| annotator | ID of the annotator |
| bbox_x0, bbox_y0, bbox_x1, bbox_y1 | Coordinates of the cutout within the original virtual slide image (axis-parallel bounding box, with roughly manually defined borders) |
| shape | Type of annotation shape ("Polygon," "Circle," or "Rectangle") |
| points | Coordinates of the points of the annotation shape. For Polygon = [x0, y0, y1, y1, …], for Circle = [x, y, r], for Rectangle = [x0, y0, x1, y1, x2, y2, x3, y3] (rotated bounding box) |
| image_id | The original BIIGLE image ID |
| image_filename | The filename of the virtual slide image the annotation was cut out |
| cutout_filename | The filename of the cutout |

2-dimensional scatterplot in Fig. 4, depicting the 10 most abundant species. To generate this figure, we computed a high-dimensional feature representation for each cutout using a ViT-L/16 vision transformer model [ViT-L/16, 96] and projected this into a two -dimensional space using t-distributed stochastic neighbor embedding (t-SNE) [97]. The embedded data were visualized using a scatterplot, where species membership is indicated by the colors used. Each data point therefore depicts one cutout. In Supplementary Figure S1, an interactive 3-dimensional version is available, allowing the visualization of all 144 species represented by at least 50 examples, with the ability to hide or display certain species interactively.

## Dataset description

All the samples processed for this dataset were taken in continental rivers, streams, and lakes, and the salinity of the habitats varied from freshwater to saline. Supplementary Table S1 contains the sampling metadata for the 319 virtual slides from which the image cutouts were generated. Table 2 contains information on the annotations that are included in the dataset as comma-separated fields (with strings quoted). The image cutouts are based on very roughly, manually annotated object shapes or rotated bounding rectangles, which usually include a substantial margin around the objects and are provided as 8-bit grayscale/intensity PNG files with a uniform resolution of 0.09 μm/pixel. The dataset contains 83,570 images of 611 diatom taxa. Of these images, 74,410 were identified at the species level to 542 species (Supplementary Table S2), the rest to 69 genera. In total, 101 species are represented by at least 100 examples each (67,594 images in total), 144 species by at least 50 examples (70,567 images in total), and 196 by at least 25 examples (72,405 images in total). The abundance distribution is highly skewed, that is, the dataset is strongly imbalanced, as typical for nonselectively collected biodiversity data (Fig. 5).

## Reuse potential

We present two deep learning experiments, each addressing particular challenges of deep learning as applied to diatom analysis. The first experiment uses a deep learning approach to handle the detection of out-of-distribution samples and explicitly models intraclass heterogeneity. Out-of-distribution detection should pinpoint specimens of taxa not present in the training set. Mod-

eling within-class heterogeneity can help to address the distinct visual appearance of valves lying in different orientations relative to the microscope view. The second experiment investigates the potential of self-supervised learning (SSL) to alleviate the need for human expertise to annotate image collections. Here, SSL utilizes unlabeled image data to learn better feature representations. The results are compared to a study conducted with a vision transformer model.

### *Deep learning experiment 1: out-of-distribution sample detection*

In this experiment, we addressed the problem of detecting out-of-distribution (OOD) samples. Deep learning classifiers often exhibit a tendency to make overconfident predictions when confronted with OOD data, erroneously classifying them as belonging to one of the classes within their training data, resulting in unreliable model outputs [98, 99]. This corresponds to a situation where a model encounters a species not represented in its training set. Instead of classifying such examples into the next best species available, it would be preferable to recognize such cases as novelties. A closely related problem is the preference of many diatom species to settle mostly at specific viewing angles on the slide (Fig. 2D) and only rarely in intermediate orientations. This leads to a discontinuous feature space, where models would need to learn to classify visually rather distinct appearances into one and the same class. This can be addressed by considering distinct views as OOD samples for other views and therefore splitting a class into visually more homogeneous clusters, which is accomplished by moving such OOD examples into appropriate own classes. In general, our OOD detection approach could enhance the reliability and safety of deep learning classifiers when facing data that deviate from their training distribution but also in cases when single classes are represented by visually distinctly different clusters of images.

For the experiments, we considered two subsets of the data. Dataset D25 included 196 classes (species) represented by at least 25 examples (individuals) as an in-distribution dataset, with the images from the remaining 346 classes used as OOD data. Dataset D50 included 144 classes represented by at least 50 examples, with the images from the remaining 398 classes being used as OOD data. For both D25 and D50, 70% of the images from the in-distribution datasets were used for training, 20% for validation, and 10% for testing.

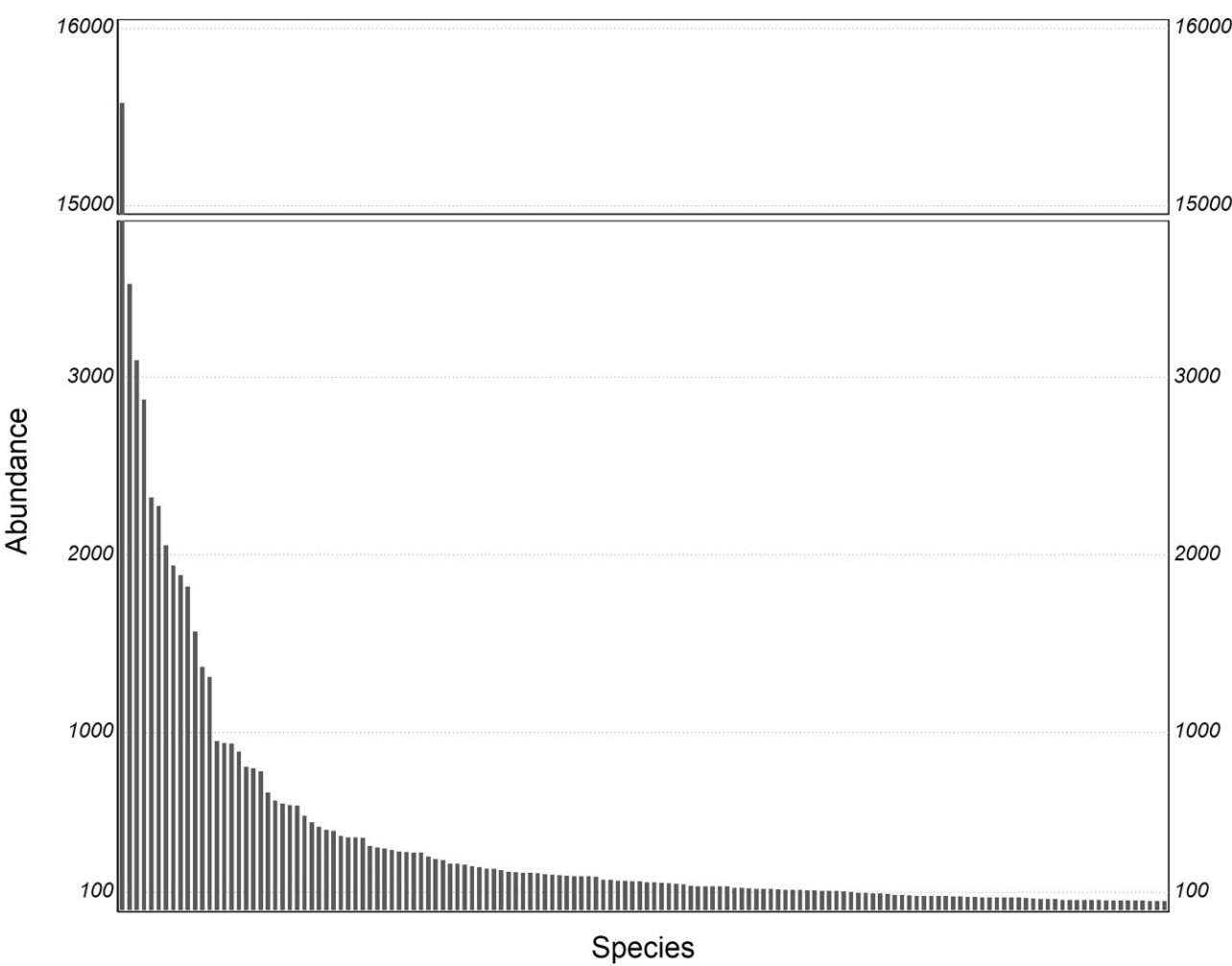

**Figure 5:** Abundance distribution of the 144 classes with at least 50 examples, illustrating the data imbalance typical of biodiversity datasets.

An EfficientNet network, pretrained on ImageNet [100], was trained using our method called MAPLE (MAhalanobis distance based uncertainty Prediction for reLiablE classification [101]) illustrated in Fig. 6. To address high intraclass variances due to, for instance, different viewpoints from which the images were acquired, we use X-means clustering [102] to break down classes into multiple clusters, each of which contains images clustering together in the feature space of representations learned by the network. These clusters are then treated as if they were different classes during the training process.

The triplet loss [103] during our training serves to bring similar samples from the same class closer together and push them farther away from samples in other classes. This approach assists the model in distinguishing between diatoms that look similar but belong to different classes.

As a baseline for comparison, the standard ImageNet-pretrained EfficientNet model trained using cross-entropy loss was used. We refer to this baseline as the deterministic counterpart of MAPLE in the results below.

Accuracy and F1-score (Table 3) were used to assess classification performance of the models (in the case of MAPLE, on in-distribution data). In addition, we used area under the receiver operating characteristic curve (AUROC) and area under the precision–recall curve (AUPR) scores for evaluation of OOD sample detection in the experiment, following common practice in the OOD literature [104–106]. The AUROC metric measures the model's ability to distinguish between in-distribution and out-of-distribution instances across various decision threshold settings. Similarly, the AUPR metric emphasizes the model's ability to perform well in situations with class imbalance. In the case of the deterministic baseline, we used the probabilities from the softmax values and, in the case of MAPLE, the probability derived from Mahalanobis distance.

Although accuracy of the deterministic model was marginally better, MAPLE achieved a higher AUROC and AUPR score compared to the deterministic classifier for both the D25 and the D50 datasets (Table 3 and Fig. 7). This outcome signifies that MAPLE demonstrates superior performance in terms of OOD sample detection.

Figure 8 illustrates subclusters found within individual species by MAPLE, which often correspond to morphologically interpretable visual differences: for instance, pleural vs. valvar views (Fig. 8A, B) in *Achnanthidium atomoides*, or single vs. both valves in *Amphora pediculus* (Fig. 8C, D). In some cases (e.g., *Fragilaria pectinalis*), different subclusters contain what seem to represent different phases of a size reduction series (Fig. 8E, F). It is unclear if this might be an artifact of having sampled two relatively distinct parts of a morphological continuum or caused by the fact that visual variation along the size axis is so much larger than in other directions. These aspects merit further investigation.

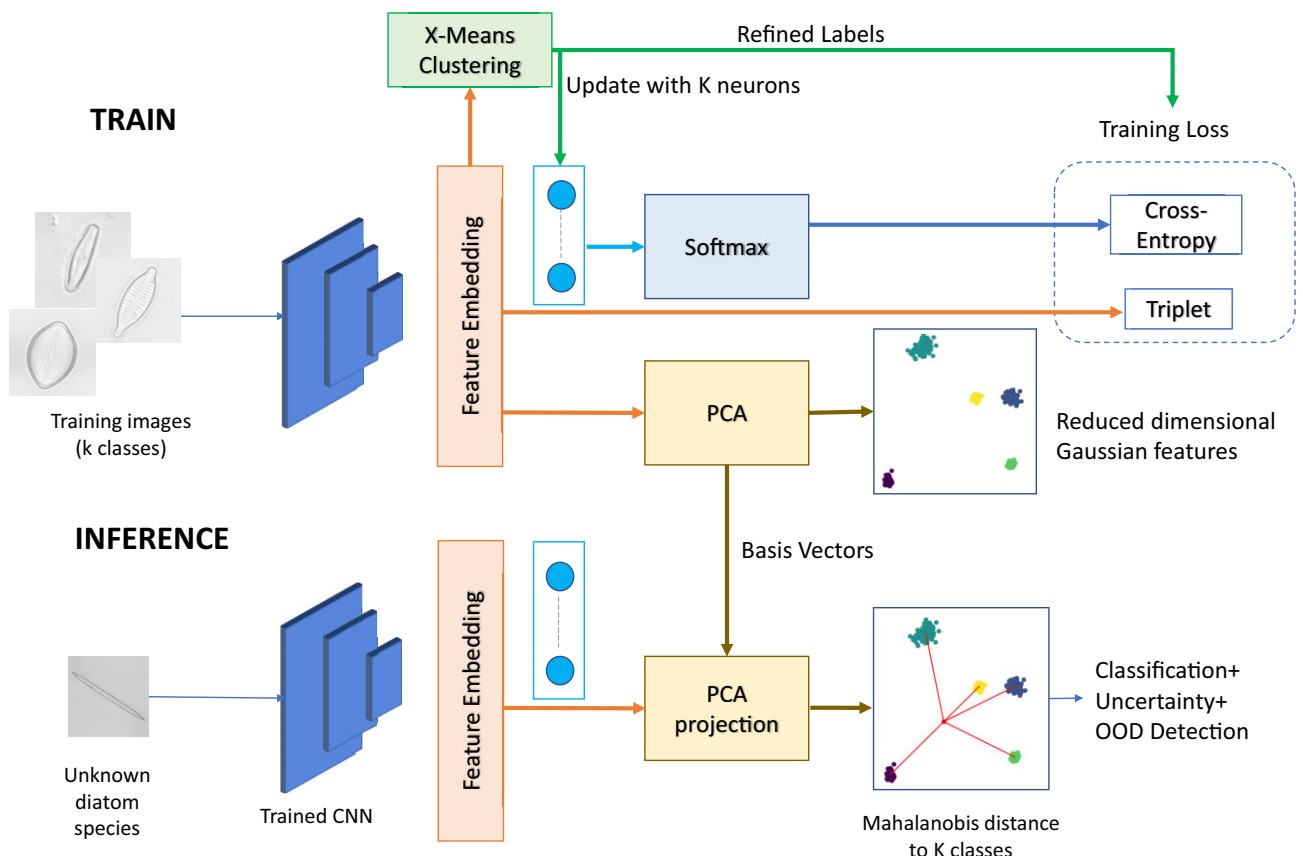

**Figure 6:** Pipeline for OOD sample detection in diatoms using MAPLE (experiment 1). During training, heterogeneous classes are split into subclasses by X-means clustering, resulting in refined labels (corresponding to these subclasses/clusters). A triplet loss supports separation of classes. During inference, a principal component analysis (PCA) projection learned during the training phase is applied to feature embeddings and is used as input for a Mahalanobis distance–based uncertainty quantification and OOD sample detection.

**Table 3:** Evaluation metrics for the OOD sample detection experiment (experiment 1). For a given dataset, a metric score in bold is higher when comparing deterministic and MAPLE methods.

| Dataset | Method | Accuracy | F1-score | AUROC | AUPR |
|---|---|---|---|---|---|
| D25 | Deterministic | **72.60%** | 0.5622 | 0.8046 | 0.8243 |
| D25 | MAPLE | 71.75% | 0.5610 | **0.8388** | **0.8421** |
| D50 | Deterministic | 60.41% | 0.5639 | 0.6844 | 0.6618 |
| D50 | MAPLE | **76.65%** | 0.5531 | **0.7282** | **0.7145** |

## Deep learning experiment 2: self-supervised learning

In our second set of experiments, we examined the impact of SSL on diatom classification. SSL is a methodology to improve classification performance by using unlabeled data [100, 107–111]. The basic idea is that prior to training the classifier in the usual supervised way, a so-called pretext task is learned. This pretext task may be, for example, as in our case, to re-create parts of the image that have previously been randomly masked (i.e., restore the full image from a version where portions of it had been deleted). For these tasks, no label information is necessary, which is why it is called self-supervised learning. During the pretext task, the algorithm learns a representation of the data in general. These representations are technically the same as a pretrained model (i.e., weights that are loaded by the algorithm, just like the usually used ImageNet pretrained models). In the SSL experiments,

we used the 144 classes from the UDE Diatoms in the Wild 2024 data that contained a minimum of 50 examples (DS50 dataset, as in experiment 1). This dataset was divided into a training set called $D^t$ (80%) and a test set called $D^{test}$ (20%). Furthermore, we randomly selected 10% of the data from each class in $D^t$ as the reduced training subset, named $D^t_{0.1}$, to simulate a scenario where training data were limited and to study the impact of SSL in this case. The structure of the datasets is illustrated in Fig. 9.

The workflow of our experiments is displayed in Fig. 10. To establish a baseline, we used a ResNet50 (hereafter referred to as RN for brevity) convolutional neural network and a ViT-Large (ViT-L/16, hereafter referred to as ViT for brevity) [ViT-L/16, 86] vision transformer model that had been pretrained on ImageNet[92] data, fine-tuned it on $D^t$, and evaluated it on $D^{test}$. These experiments are referred to as $RN_{D^t}$ and $ViT_{D^t}$, respectively. We conducted

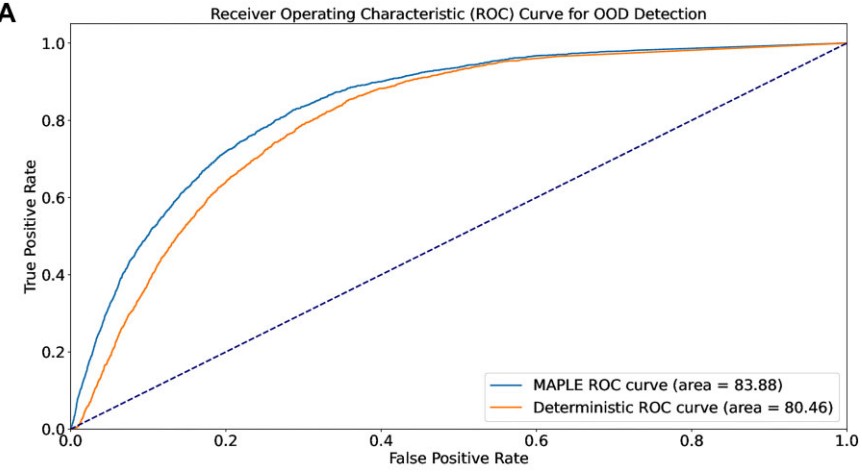

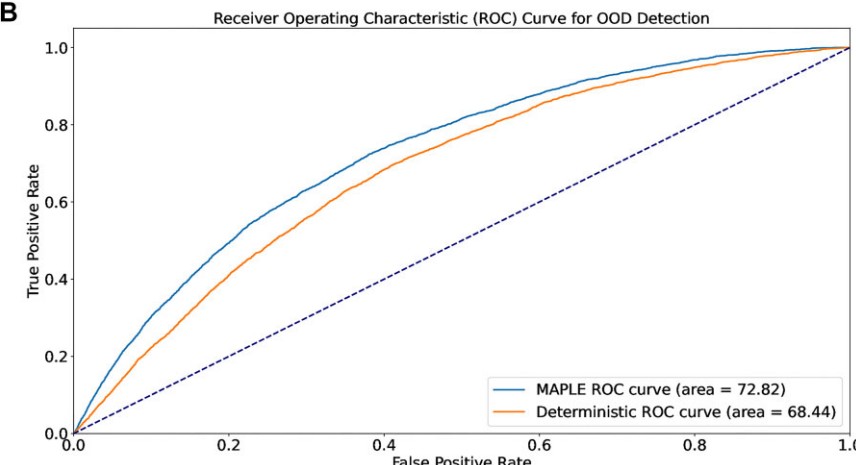

**Figure 7:** Receiver operating characteristic curves from the OOD sample detection experiment (experiment 1) for the deterministic vs. MAPLE methods on the D50 and on the D25 datasets.

identical experiments, using the smaller $D_{0.1}^t$ training data subset, referring to them as $RN_{D_{0.1}^t}$ and $ViT_{D_{0.1}^t}$, respectively.

To compare a self-supervised approach with the ViT baseline, we employed a masked auto-encoder (MAE) [100] using the same back-end ViT. This MAE had already been pretrained using SSL on ImageNet data, and we fine-tuned it on $D^t$. In this case, nondomain data were used for SSL training, but the fine-tuning was done on in-domain data. These experiments are denoted as $MAE_{D^t}$ and $MAE_{D_{0.1}^t}$.

The results of the experiment showed that network performance benefited from SSL, whether fine-tuned with the whole labeled dataset $D^t$ or with only 10% of the labeled data $D_{0.1}^t$, as measured both by macro- and micro-averaged metrics [112] (Table 4).

### Conclusion from deep learning experiments

Our results reached substantially lower accuracies, in comparison to deep learning experiments previously applied to diatom data [45]. We attribute this to our nonselective imaging method, which impacts the specimen and image quality as well as the background homogeneity, and also has an effect on the intra- and interclass variations of features, all of which probably make our "UDE Diatoms in the Wild 2024" dataset more challenging but also reflective of a typical use case. As discussed in the introduction,

this is by design: we think it is important to apply and test image analysis methods on types of image data that can be produced by high-throughput imaging methods, as opposed to manual selection and focusing by a human expert.

Beyond its relevance to diatom analysis and more broadly to biodiversity and environmental research, this dataset is demanding also from a general computer vision point of view. Unlike previously available "clean" datasets, which are typically used as benchmarks in the computer vision community, this dataset contains several of the problems typically encountered when dealing with real-life datasets. This includes a class imbalance, resulting in a long-tailed distribution of the images for classification. Such class imbalances pose difficulties for machine learning approaches as the overrepresented classes have a stronger influence on the acquired model. Additionally, the dataset exhibits high levels of interclass similarity and intraclass variance due to the special visual features of diatoms outlined in the introduction. Moreover, the presence of occlusions (diatoms being partly concealed by overlapping objects) within the dataset adds another layer of complexity. Dealing with occlusions requires robust feature extraction and recognition capabilities to effectively discern obscured objects. Some of these problems are not unique to diatom classification but are general problems investigated by computer vision for decades now. Given these listed observations, this

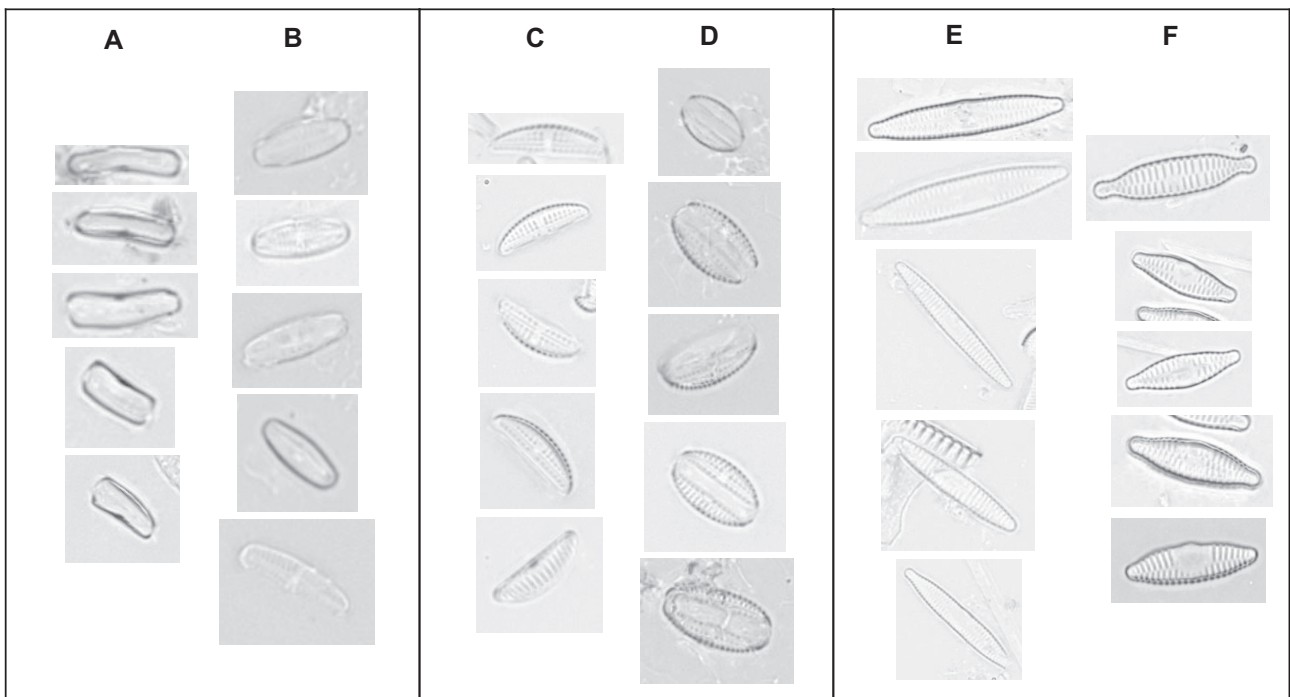

**Figure 8:** Examples illustrating subclusters within individual species delimited by MAPLE. (A, B) *Achnanthidium atomoides* in pleural (A) vs. valvar view (B). (C, D) *Amphora pediculus*, represented as single valve (C) vs. both valves together (D). (E, F) Subclusters in *Fragilaria pectinalis* appear to depict life cycle–associated variants.

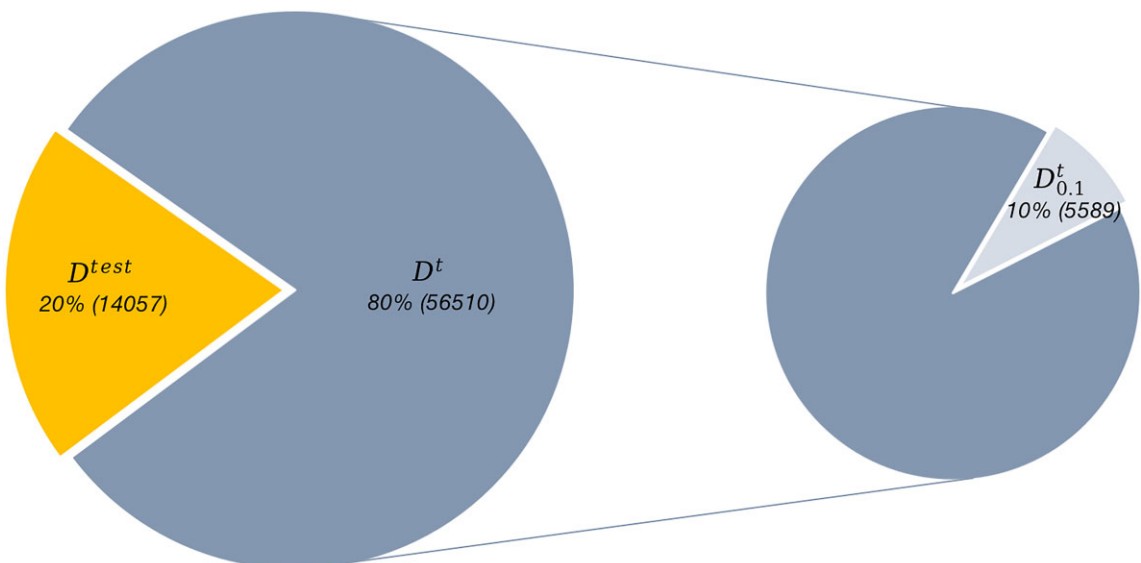

**Figure 9:** Structure of datasets for experiment 2. Twenty percent of the images were used as a test set ($D^{test}$). In 1 experiment, all the remaining (80%) images were used for model training (denoted $D^t$ on the left-hand side). In a second experiment, only 10% of training data of each class in $D^t$ (denoted $D^t_{0.1}$ on the right-hand side) was used for model training to investigate the effect of dataset size.

dataset can be seen as a valuable resource not only for diatom research but also for addressing some more generic challenges in computer vision.

We would also argue that for applicability of digital imaging and identification methods for routine diatom community characterization or, for instance, water quality monitoring, intelligent combinations of advanced models (going beyond simple supervised classification, like our baseline models) will be necessary.

For instance, as experiment 2 shows, self-supervised learning has a potential to alleviate the need for labeled training data, whereas out-of-distribution detection, as possible in MAPLE (experiment 1), has the potential to address detecting taxa not represented in a training set, another practically relevant aspect of real-life analyses. How to best combine these strengths to a best overall digital diatom community analysis workflow is currently an open question.

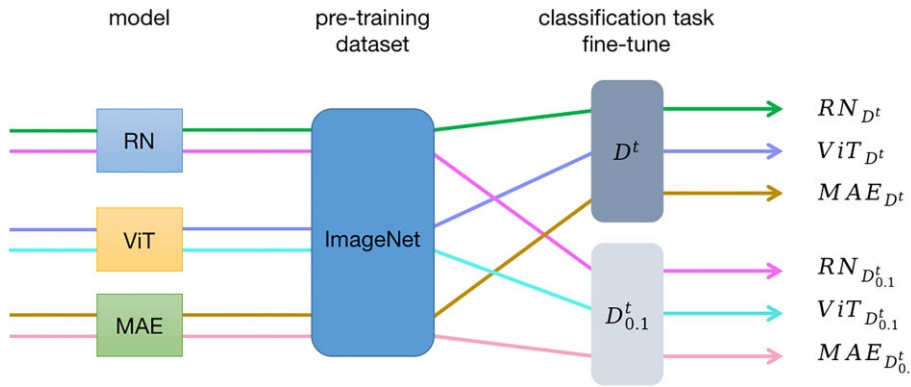

**Figure 10:** Flowchart of experiment 2. Pretraining refers to a supervised training for the baseline model (ViT) and a pretext training for the self-supervised model (MAE). Finally, all models were fine-tuned in a supervised fashion.

**Table 4:** Evaluation metrics for experiment 2. A metric score in bold is higher when comparing ResNet50 (referred to as RN), ViT, and MAE methods.

| Experiment | Macro-average accuracy | Micro-average accuracy | Macro-average F1-score | Macro-average AUROC score |
|---|---|---|---|---|
| $RN_{D^t}$ | 63.76% | 78.78% | 0.6507 | 0.9798 |
| $ViT_{D^t}$ | 60.31% | 78.04% | 0.6283 | 0.9490 |
| $MAE_{D^t}$ | **66.37%** | **80.61%** | **0.6824** | **0.9848** |
| $RN_{D^t_{0.1}}$ | 41.78% | 70.04% | 0.4315 | 0.9421 |
| $ViT_{D^t_{0.1}}$ | 42.19% | 69.97% | 0.4456 | 0.9397 |
| $MAE_{D^t_{0.1}}$ | **47.75%** | **73.22%** | **0.4941** | **0.9821** |

## Abbreviations

AUPR: area under the precision–recall curve; AUROC: area under the receiver operating characteristic curve; MAE: masked autoencoder; OOD: out of distribution; SSL: self-supervised learning; t-SNE: t-distributed stochastic neighbor embedding.

## Availability of Source Code and Requirements

Project name: UDE Diatoms in the Wild—experiment 1: MAPLE out-of-distribution detection
Project homepage: https://github.com/vaishwarya96/maple-ude
Operating system(s): LINUX
Programming language: Python 3
Other requirements: described in the GitHub repository
License: MIT License
Project name: UDE Diatoms in the Wild—experiment 2: self-supervised learning
Project homepage: https://github.com/mtan-unibie/GIGASci
Operating system(s): LINUX
Programming language: Python 3
Other requirements: described in the GitHub repository
License: MIT License

## Additional Files

**Supplementary Fig. S1.** tSNE.html.
**Supplementary Table S1.** Metadata samples and slides.xlxs.
**Supplementary Table S2.** Species abundance.docx.

## Author Contributions

Aishwarya Venkataramanan (Conceptualization [supporting], Formal analysis [equal], Visualization [equal], Writing – original draft [equal], Writing—review & editing [equal]), Michael Kloster (Conceptualization [equal], Data curation [lead], Visualization [equal], Writing – original draft [equal], Writing—review & editing [equal]), Andrea Burfeid-Castellanos (Data curation [lead], Writing – original draft [equal], Writing—review & editing [equal]), Mimoza Dani (Data curation [supporting], Writing – original draft [supporting], Writing—review & editing [supporting]), Ntambwe A. S. Mayombo (Data curation [equal], Writing – original draft [supporting], Writing—review & editing [supporting]), Danijela Vidakovic (Data curation [equal], Funding acquisition [equal], Writing – original draft [supporting], Writing—review & editing [supporting]), Daniel Langenkämper (Conceptualization [equal], Formal analysis [equal], Supervision [equal], Writing – original draft [equal], Writing—review & editing [equal]), Mingkun Tan (Formal analysis [equal], Visualization [equal], Writing – original draft [equal], Writing—review & editing [equal]), Cedric Pradalier (Conceptualization [equal], Funding acquisition [equal], Supervision [equal], Writing – original draft [equal], Writing—review & editing [equal]), Tim Nattkemper (Conceptualization [equal], Funding acquisition [equal], Supervision [equal], Writing – original draft [equal], Writing—review & editing [equal]), and Martin Laviale (Conceptualization [equal], Funding acquisition [equal], Supervision [equal], Writing – original draft [equal], Writing—review & editing [equal]), Bánk Beszteri (Conceptualization [equal], Funding acquisition [equal], Supervision [equal], Writing – original draft [lead], Writing—review & editing [equal]).

## Funding

M.K. and D.L. were funded by the Deutsche Forschungsgemeinschaft (DFG, German Research Foundation; project number: 463,395,318). M.D., A.B.C., and N.A.S.M. were partially funded by the Collaborative Research Centre 1439 RESIST (Multilevel Response to Stressor Increase and Decrease in Stream Ecosystems; www.sfb-resist.de), funded by the DFG (CRC 1439/1, project number: 426,547,801). A.B.C. was also partially supported by the EU through the PRIMA project (INWAT 201980E121), which was sponsored by the German Federal Ministry of Education and Research. Funding for D.V. was provided by the Humboldt Foundation. The PhD scholarship for A.V. was funded by ANR, France (ANR-20-THIA-0010), and Région Grand-Est, France. Additional financial support was provided by CNRS, France (ZAM LTSER Moselle), and Horizon Europe (iMagine—Grant agreement ID: 101,058,625). This publication was supported by the University of Duisburg–Essen Open Access Publication Fund.

## Data Availability

Supporting data are available from the *GigaScience* database, GigaDB [113], with the full dataset deposited in Zenodo [80]. For easy practical application, subsets containing training, validation, and test data (60%/20%/20% split) of species represented by at least 25, 50, or 100 specimens each and stored in the simple torchvision DatasetFolder-dataset structure with one folder per species are available from Kaggle [81].

## Competing Interests

The authors declare no competing interests.

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
