## [Peer Review File · GigaScience]

Author's Response To Reviewer Comments

Replies included in uploaded document. Also copy-pasting below:

Reviewer reports:

Reviewer #1: "UDE DIATOMS in the Wild 2024": A new image dataset of freshwater diatoms for training deep learning models

General Comments:

The rationale provided for the purpose of the dataset is to improve the objective, reproducible and comparable nature of diatom data. The components of the problem seems to be 1) obtaining images, and what is done to them - annotation and resolving differing annotations; and 2) evaluating what deep learning seems to produce from the images. Notably, the images are not idealized, but have all of the complications of specimens encounter by human microscopists.

The abstract uses "findings" which does not seem to fit the context. Similarly, "conclusions" could be stating the informative outcomes of the two experiments.

Thank you for these comments. The abstract sections follow the journal guidelines, so we chose to leave them as is. As for the conclusions to state the results of the experiments, the Author instruction states that "Conclusions: a short summary of the potential uses of these data and implications for the field.", which is the way it is formulated. Considering also the word count limit, we thus also here wdecided to stay with our original formulation.

I am not a specialist in deep learning, but a diatom biologist. Therefore, I am able to comment on the more non-technical aspects of the manuscript. In particular, I note words and phrases that could improve clarity of the text. The authors might consider providing additional use cases for those interested in obtaining the data.

The manuscript is appropriate for publication in Gigascience, pending minor revisions.

Specific Comments:

Line 24. "need for digitalisation of these methods has long been recognized". I was not aware of the term digitalization and its meaning. Consider adding the definition. Wording would be is vague - is it really the methods that are digitalized? Or by "methods", do you mean to include image capture and AI recognition?

We thought that digitalization was a commonplace term, but we replaced it by "digital transformation"; in this case, it includes finding digital counterparts to methodological steps earlier undertaken without digital representations, which includes imaging and downstream image processing in the digital case, but which were performed by manual microscopy and expert analysis of the visual information thus obtained in the non-digital case.

Line 25. One might argue that diatoms lack morphological diversity. Compared to other organisms, diatoms tend to be identified by subtle shape features; they are lacking in the number of morphological features they possess.

We disagree; of course the question is what "other organisms" to compare with, but compared to many other groups of microscopic organisms (e.g., prokaryotes, nanoflagellates), they can be considered rich in morphological features.

Line 31. As a dataset, "findings" is not an outcome of the product. Consider "data descriptor" or alternate term.

As mentioned above, "Findings" is requested by the journal as a section heading in the abstract.

Line 39. Similarly, the text here does not represent "conclusions", but a restatement of the problem.

Also as referred to above, we chose to stick with journal instructions.

Line 62. Omit "these" in "A need for digitalisation of light microscopic methods". So you include "identification" in the method?

Yes, we do, we now specified this, the modified sentence: "A need for a digital transformation of these light microscopic and manual identification methods..."

Line 68. "performed directly on a microscope" - Do you mean "performed by microscopists" or "performed by diatomists, using a microscope"?

Clarified as "identifications performed by human experts directly on a microscope".

Line 70. Consider rewording as, "Over 20 years ago, the ADIAC project developed fundamental approaches for digital imaging and identification [24]. More than ever, we now need standardized, digital imaging methods combined with automated (?) taxonomic identification in order to have objective, reproducible, and comparable taxonomic data for rapid processing of large numbers of samples".

Thanks for the suggestion, we accepted it, with the small change that instead of "automatic", we took "digitally supported taxonomic identification" since it can be, but doesn't have to be, conceived as "automatic".

Line 84. Suggested edit to, "solved by focusing up and down through the three-dimensional structure of a valve, termed 'optical dissection'".

After a web search, we have the impression that "optical dissection" is mainly being used in neuro-imaging in a quite specific meaning that looks different from what we mean here. For this reason, we would opt to not include the term here.

Line 89. Note that some software applications (at least Olympus software) are able to compare the degree of blur in a stack and save the stratum with the sharpest focus.

Certainly; what the text refers to here is, however, that these algorithmically chosen "ideal / sharpest" planes do not necessarily conform to an ideal plane for a human expert that tries to identify a diatom, and that for the latter, no generally applicable objective criteria are available (for instance, when the valve edge is sharpest, usually the valve surface ornamentation is out of focus and vice versa – which of both is more informative for identification, and which contributes more information for an algorithmic autofocus, might be different for different taxa or even individual diatom objects). We would not like to go into so much detail for such a side aspect in the introduction, so we opted for a more compact modification of the sentence as "the problem of finding the optimal focal plane for taxonomic identification of each diatom specimen automatically...".

Line 100. What is the difference between a gigapixel-sized slide scan or a virtual slides? Clarify, as these seem like the same thing to me.

Yes, we intended to use them synonymously; corrected to make this clearer to "gigapixel-sized slide scans sometimes termed "virtual slides"".

Line 103. Replace, "There are a number of factors" with "Several factors"

Changed.

Line 105. Edit to "local or regional"

Changed.

Line 111. Omit "highly", as something is either time-consuming or not.

We disagree – we see "time-consuming" on a continuous, rather than a binary, scale.

Line 134. Omit "highly"

Changed.

Line 136. Consider edit to, "Human analysts learn to interpret and link differing orientations with experience"

Changed.

Line 159. Consider edit to, "Thus, analyses of light microscopic images of diatoms by deep learning is an urgent need for research of ecology and biodiversity, as well as environmental monitoring. Yet, development of the machine learning and computer vision is a challenge.

Changed.

Line 163. What is "benchmarking"? Not being a specialist, I need clarification. OR could this sentence be just as meaningful by stating, "datasets that are suitable for training deep learning models". What is the importance of benchmarking?

Changed to "comparing".

Line 163. "There are very few extensive taxonomically annotated diatom image datasets publicly available to begin with, and the available ones are mostly too small to be suitable for training deep learning models." This would be more clearly written as,

"Few diatom image datasets have associated taxonomic annotations, and of those that do have annotations, the datasets are generally too small to be suitable for training deep learning models."

Changed in a slightly different way: "There are very few publicly available diatom image datasets, and the available ones are mostly too small for training deep learning models."

Line 167. "pre-deep learning machine" is sort of a weird phrase.

Changed to "a machine learning utilization before the deep learning era".

Line 177. Cite diatoms.org as:

Spaulding, S.A., Potapova, M.G., Bishop, I.W., Lee, S.S., Gasperak, T.S., Jovanovska, E., and Edlund, M.B. 2021. Diatoms.org: supporting taxonomists, connecting communities. *Diatom Research* 36(4): 291-304. doi:10.1080/0269249X.2021.2006790

Added citation.

Line 180. Cite Kaggle as:

<https://www.kaggle.com/> - unless they have an alternate citation.

Added.

Line 197. "is being made publicly available to support customizing and benchmarking deep learning models to this field of application."

Is it available now? Can you state, "is publicly available to support customizing and benchmarking deep learning models for diatoms".

Changed.

Line 290. I would think the header should be "Potential uses"

"Re-use potential" is a section head requested by the journal.

Line 291. "Challenging nature of the dataset" sounds like the dataset is difficult to use, rather than (and I think this is what you mean), "deep learning as applied to diatoms is challenging because...." Clarify, especially because it seems that "challenging" is defined differently in the abstract. Suggested revision to place the purpose of each experiment first:

We present two deep learning experiments, each addressing a particular challenge of deep learning as applied to diatom analysis. The first experiment is useful for illustrating the situation in which specimens of taxa are encountered that were not present in the training set. We expect the experiment to demonstrate (?) the distinct visual appearance of valves lying in orientations not previously encountered (?). The first experiment uses a deep learning approach to handle the detection of out-of-distribution samples and explicitly models intra-class heterogeneity.

The second experiment is useful for investigating the potential of semi-supervised learning (SSL) to alleviate the need for human expertise to annotate image collections.

Here, SSL simulates unlabelled image data to learn better feature representations. The results are compared to a study conducted with a vision transformer model.

Thank you for the suggestions. We took the first sentence and the text for experiment 2 as suggested; the other sentences for experiment 1, however, would change the meaning, here we chose a different rewording. The whole paragraph was accordingly changed to: "We present two deep learning experiments, each addressing particular challenges of deep learning as applied to diatom analysis. The first experiment uses a deep learning approach to handle the detection of out-of-distribution samples and explicitly models intra-class heterogeneity. Out-of-distribution detection should pinpoint specimens of taxa not present in the training set. Modelling within-class heterogeneity can help to address the distinct visual appearance of valves lying in different orientations relative to the microscope view. The second experiment investigates the potential of semi-supervised learning (SSL) to alleviate the need for human expertise to annotate image collections. Here, SSL utilizes unlabelled image data to learn better feature representations. The results are compared to a study conducted with a vision transformer model."

Line 333. What is "triplet loss"?

Citation added.

Line 389. What is "DL"?

Changed to "deep learning".

Line 390. Would be better written as "Our results show x process produced lower accuracies of y,z"

This sentence refers to all experiments performed in our paper, thus the more general formulation is intended.

Line 394. Okay, now I understand why you say the dataset is challenging. But this differs from earlier definition - you can also add that the dataset is more reflective of the typical use case.

Thanks - we added this formulation to the sentence.

Line 407. What is "presence of occlusions within the dataset"?

Changed to shortly explain to: "occlusions (diatoms being partly concealed by overlapping objects)".

Line 411. Who is "computer vision community". If this is your audience, make that clear early on rather than at the end.

We don't know who they are – modified to: "Some of these problems are not unique to diatom classification, but are general problems investigated by computer vision for decades now. Given these listed observations, this dataset can be seen as a valuable resource not only for diatom research, but also for addressing some more generic challenges in computer vision."

Reviewer #2: The paper introduces a new image dataset for training and testing models for diatom recognition. The dataset is considerably larger than previous datasets and provides a more realistic benchmark for diatom recognition methods than earlier 'clean' datasets. Due to the fine-grained nature of the diatom recognition task, the dataset has the potential to be useful in deep learning model development beyond the application area. Therefore, the dataset is definitely worth publishing. However, due to the shortcomings in the manuscript, I cannot recommend its publication in its current form.

The main comments:

1) The set of experiments included in the paper is somewhat unusual for a dataset paper. I would expect to see baseline experiments of common (closed-set) recognition models (ResNet, ViT, etc.) with the entire dataset. Instead, the authors provide two rather specific experiments; one on out-of-distribution detection and one on self-supervised learning. While these are both interesting experiments, it is unclear why they were selected and leave the purpose of the dataset a bit unclear. I would recommend including baseline plankton recognition experiments.

Experiment 1 included a baseline comparison (EfficientNet). Reflecting the critique of reviewer 2, we now also added two baselines (ResNet50 and ViT/L-16) to experiment 2. The purpose of the advanced models is described in the description of the experiments as well as in the Conclusion in detail; to complement, we added a sentence to the end of the introduction as: "To highlight the challenging nature of this dataset, as well as to propose possible avenues to address some of these challenges, we provide two deep learning experiments, one addressing out-of-distribution detection and modelling within-class heterogeneity, another one leveraging semi-supervised learning to alleviate the need for voluminous labelled training data."

2) The paper lacks a clearly defined evaluation protocol. One of the main problems with most existing diatom datasets, and plankton recognition datasets in general, is that there is no standardized way to use them. This leads to a situation where different papers apply the data differently (different splits, evaluation metrics, etc.), and the results in different papers are not comparable despite the use of the same dataset. The paper would greatly benefit from a clear description of the evaluation protocol that everyone using the dataset could follow and replicate the experiments with their method.

We have double-checked the transparency of the evaluations and extended at a few places. In addition, since all analysis code is available incl. in containerized form, incl. seeds used to generate data splits, we believe that we have a clear, transparent and reproducible documentation of the evaluation protocols for both experiments (see also replies to points 6-8 below, which also relate to aspects of this general comment). We would also like to note besides that although all analyses presented can be seen as "simple classification models" and can be characterized as such (accuracy, F1 etc., as done in the manuscript), our goal was not to identify the best performing such model for the task of classification of this dataset, but also to draw attention to the fact that a well-performing classifier is not all that will be needed for the real-life application of digital diatom analysis methods. To make this clearer, we added to the Conclusion: "We would also argue that for applicability of digital imaging and identification methods for routine diatom community characterization or for instance water quality monitoring, intelligent combinations of advanced models (going beyond simple supervised classification, like our baseline models) will be necessary. For instance, as experiment 2 shows, semi-supervised learning has a potential to alleviate the need for labelled

training data; whereas out-of-distribution detection, as possible in MAPLE (experiment 1), has the potential to address detecting taxa not represented in a training set, another practically relevant aspect of real life analyses. How to best combine these strengths to a best overall digital diatom community analysis workflow, is currently an open question.”.

Detailed comments

3) In Table 1, the authors list existing image datasets that consist of only diatoms. While the table appears comprehensive, there is also a large pool of more general phytoplankton datasets that contain various diatom taxa. Some of these could also be listed. For example, see:

Eerola, T., et al. (2024). Survey of automatic plankton image recognition: Challenges, existing solutions, and future perspectives. *Artificial Intelligence Review*, 57(5), 114.

The datasets listed there come from plankton, whereas our targeted habitat is microphytobenthos or periphyton. For this reason, and to avoid redundancy with the mentioned review, we added a citation of the reference to the introduction along with a sentence: “We note that for planktonic organisms, a much larger collection of datasets is publicly available, these were recently reviewed [76].”

4) More details on the annotation process should be provided. How many annotators in total? How many annotators per image? Was label uncertainty/conflicting expert labels addressed in some way? In lines 257-258, the authors state that 'The annotations were ... filtered to remove irrelevant labels and annotations from inexperienced annotators.' What does this mean?

Information added in the Annotation section (4 annotators; 1 annotator / image; thus, no conflicts). The mention of the filtering step was removed since we realised that it is not relevant for describing the data set (some non-taxonomic labels were for instance used for within-group exchange, these were removed).

5) The term 'out-of-distribution detection' is commonly used but a bit confusing in this context because the word 'detection' is used with two meanings in the paper (the other being the localization of the diatom in an image). Perhaps the term 'out-of-distribution recognition' could be used instead.

We checked once more all occurrences of the word “detection” in the text. We have only two occurrences in the introduction where it appears in a different meaning, but also clearly differentiated as “object detection”. In all other appearances, it is clearly and explicitly used as “out-of-distribution detection”. Since, as the reviewer notes, “out-of-distribution detection” is a standard term, and we did not want to come up with a new name for it, we decided to leave this formulation.

6) It seems that the OOD detection experiment does not include non-diatom particles. As the original images contained background particles (sediment, clay, diatom fragments, etc.), it would have been beneficial to include them as OOD samples. This would provide a more realistic experiment setup.

We agree with the reviewer that detecting non-diatoms would also be a relevant use case for OOD. Our main use case we had in mind for these experiments was detecting “novel” diatom taxa not represented in the training set. Since we did not collect image annotations for non-diatom particles, we cannot address this aspect in the present paper.

7) It is unclear how the baseline method is applied in the OOD experiment. Is it thresholding of softmax probabilities? Training a binary classifier with the OOD samples in the training set?

See next question.

8) I don't think AUROC and AUPR are very good evaluation metrics as the numbers are very hard to interpret (what does an AUROC of 0.8388 actually mean in practice?). Typically, we want to find the best decision threshold and are interested in classification accuracy with that threshold.

We reply together to these two points because we addressed them by changing the text describing these

aspects of experiment 1 to make this clearer. No binary classifier was trained, but uncertainty evaluation was indeed done based on the softmax probabilities. In this aspect, as well as in taking AUROC and AUPR as central measures to characterize the out-of-distribution detection performance of the models, we followed common practice in the OOD literature; we now added references. As for selecting a threshold, this would have been quite arbitrary; also in general, different use cases can warrant different preferences in terms of the trade-off between e.g. precision and recall and accordingly, different threshold selection. Although we realise that AUROC or AUPR is difficult to interpret intuitively, their main task here is to allow a comparison in the OOD detection performance of MAPLE vs. a baseline model, which they allow quite simply. The modified text: "Accuracy and F1-score (Table 3) were used to assess classification performance of the models (in the case of MAPLE, for in-distribution data). In addition, we used AUROC (area under the receiver operating characteristic curve) and AUPR (area under the precision-recall curve) scores for evaluation of OOD sample detection in the experiment, following common practice in the OOD literature [93-95]. The AUROC metric measures the model's ability to distinguish between in-distribution and out-of-distribution instances across various decision threshold settings. Similarly, the AUPR metric emphasizes the model's ability to perform well in situations with class imbalance. In the case of the deterministic baseline, we used the probabilities from the softmax values, and in the case of MAPLE, the probability derived from Mahalanobis distance."

9) The abbreviation SSL is used with two different meanings: semi-supervised learning (e.g., page 14) and self-supervised learning (e.g., page 16). The same applies to the description of what the second experiment is about. While self-supervised learning can be utilized in semi-supervised learning, they are definitely not the same thing.

Thanks for pointing this out – changed all occurrences to "semi-supervised". "Self-supervised" can be used to characterize the pretext task used as a step of the procedure, but responding to the question we decided to limit our text to only using one of the two terms to avoid confusion.

10) What are micro- and macro-average in Table 4?"

See <https://doi.org/10.1016/j.ipm.2009.03.002> - we added a citation to the text.